# Superplasticity in a lean Fe-Mn-Al steel

Jeongho Han[1,4], Seok-Hyeon Kang[1], Seung-Joon Lee[1,5], Megumi Kawasaki[2], Han-Joo Lee[2], Dirk Ponge[3], Dierk Raabe[3] & Young-Kook Lee[1]

Superplastic alloys exhibit extremely high ductility (>300%) without cracks when tensile-strained at temperatures above half of their melting point. Superplasticity, which resembles the flow behavior of honey, is caused by grain boundary sliding in metals. Although several non-ferrous and ferrous superplastic alloys are reported, their practical applications are limited due to high material cost, low strength after forming, high deformation temperature, and complicated fabrication process. Here we introduce a new compositionally lean (Fe-6.6Mn-2.3Al, wt.%) superplastic medium Mn steel that resolves these limitations. The medium Mn steel is characterized by ultrafine grains, low material costs, simple fabrication, i.e., conventional hot and cold rolling, low deformation temperature (ca. 650 °C) and superior ductility above 1300% at 850 °C. We suggest that this ultrafine-grained medium Mn steel may accelerate the commercialization of superplastic ferrous alloys.

[1] Department of Materials Science and Engineering, Yonsei University, Seoul 03722, Republic of Korea. [2] Division of Materials Science and Engineering, Hanyang University, Seoul 04763, Republic of Korea. [3] Max-Planck-Institut für Eisenforschung, Max-Planck-Straβe 1, Düsseldorf 40237, Germany. [4] Present address: Department of Materials Science and Engineering, Chungnam National University, Daejeon 34134, Republic of Korea. [5] Present address: Joining and Welding Research Institute, Osaka University, Osaka 567-0047, Japan. Jeongho Han and Seok-Hyeon Kang contributed equally to this work. Correspondence and requests for materials should be addressed to Y.-K.L. (email: yklee@yonsei.ac.kr)

Superplasticity is defined as ultrahigh ductility above 300% without cracking, which is primarily achieved by grain boundary sliding (GBS). It is promoted by fine grain size, low strain rate, high deformation temperature above half of melting point[1–3]. The extreme ductility of superplastic alloys enables forming parts with nearly arbitrarily complex shapes. Until now, most studies on superplasticity were conducted on non-ferrous materials, such as Zn[4–6], Ni[7, 8], Al[9, 10], and Ti[11, 12] alloys. Although these compounds revealed good superplastic behavior, they are not suited for mass production and high strength applications due to their high material costs (Ni and Ti alloys) and insufficient strength after superplastic forming (Zn and Al alloys). To overcome these limitations, iron-based superplastic alloys, such as duplex stainless steel (DSS)[13–17] and ultrahigh carbon steel (UCS)[18–22], were developed. Although showing promising superplasticity, these steels are not applied commercially due to their high alloying contents (Cr, Ni, and Mo), high material costs and high deformation temperature (>800 °C) in the case of DSS and complicated fabrication process in the case of UCS, respectively.

In recent years, Fe-medium Mn steels (MMSs) with less than ~10 wt.% Mn and ~0.3 wt.% C have been actively investigated as the next-generation automotive steels due to their low material costs and excellent mechanical properties[23–26]. The alloys have a homogeneous composite microstructure consisting of α ferrite and γ austenite after cold rolling and intercritical annealing at temperatures of ~600–900 °C. Both phases have ultrafine equiaxed grains, ranging from ~300 nm to 3 μm in diameter. Because the fine grain size (less than or equal to ~10 μm) is known to be a prerequisite for superplasticity, ultrafine-grained MMSs have high potential for showing excellent superplastic ductility. Nevertheless, until now only DSSs and UCSs have been studied as iron-based superplastic materials[13–22].

Here we propose and develop a new superplastic MMS that is suitable for the mass production of parts with complex shapes and high strength. The superplastic MMS with a dual-phase microstructure of α ferrite and γ austenite has characteristics of low material cost, simple fabrication process, remarkable total elongation (>1300%), low deformation temperature (650 °C and above) and high strain rate above $1.0 \times 10^{-3}$ s$^{-1}$. In addition, the MMS reveals significant dislocation hardening even at high strains unlike superplastic DSS with the same dual-phase microstructure.

## Results

**Alloy design.** In the present study the superplasticity of an Al-bearing MMS was investigated using Fe-6.6Mn-2.3Al-0.005 C (wt.%) cold-rolled steel. Al was added to increase the temperature range of the α plus γ dual-phase superplastic flow regime, where the high-temperature tensile tests were performed. The approach of adding Al for widening the dual-phase region into the forming region of interest stemmed from preliminary experiments using an Al-free Fe-7Mn-0.05 C (wt.%) steel. The steel exhibited an elongation of ~270% at 645 °C at an initial strain rate of $1.0 \times 10^{-4}$ s$^{-1}$ (Supplementary Fig. 1), which is insufficient for superplasticity most likely due to the low tensile testing temperature.

**Superplastic tensile behavior.** Figure 1a and b shows tensile-fractured Fe-6.6Mn-2.3Al-0.005C (wt.%) specimens and the associated engineering stress–strain curves, respectively. The tensile specimens were heated at ~0.5 °C s$^{-1}$ to 850 °C, where the equilibrium volume fractions of both α and γ phases are almost identical, held for 5 min for temperature homogeneity and γ formation, and then tensile-deformed at various initial strain rates. All specimens exhibited superplastic flow with elongations in excess of 300% regardless of strain rate. A maximum elongation of ~1314% was obtained for an initial strain rate of $1.0 \times 10^{-3}$ s$^{-1}$. It is well known that when the strain rate sensitivity ($m = \partial \ln\sigma / \partial \ln\dot{\epsilon}$) exceeds ~0.3, strain localization and localized necking (but not diffuse necking) are suppressed during plastic deformation and GBS occurs, resulting in superplasticity[2, 11, 27]. Accordingly, the $m$ value of the present steel was determined at several true strain levels. The average $m$ ($m_{avg}$) value was above 0.3; 0.33 at the initial strain rate range from $1.0 \times 10^{-4}$ to $1.0 \times 10^{-3}$ s$^{-1}$; 0.39 from $1.0 \times 10^{-3}$ to $1.0 \times 10^{-2}$ s$^{-1}$; and 0.33 from $1.0 \times 10^{-2}$ to $1.0 \times 10^{-1}$ s$^{-1}$ (Fig. 1c). However, elongation depends not only on the $m$ value but also on the ratio of gauge length to width[28], more specific, the lower the ratio the larger the elongation. The ratio of gauge length to width in the specimens used in the present study is 0.5, which is lower than the ratio of gauge length to width (0.74–2) in the specimens used in literature[6, 8, 13, 16, 17]. Accordingly, if the ratio of gauge length to width increases, the elongation of the specimens is expected to be reduced.

Additional tensile tests were carried out at temperatures ranging from 650 to 900 °C (Supplementary Fig. 2). When the initial strain rate was $1.0 \times 10^{-3}$ s$^{-1}$, an elongation of over 400% was achieved even at a temperature as low as 650 °C. This result is fascinating, considering the fact that DSSs with α and γ phases such as the present MMS show superplasticity only at relatively high temperatures above 800 °C[13–17].

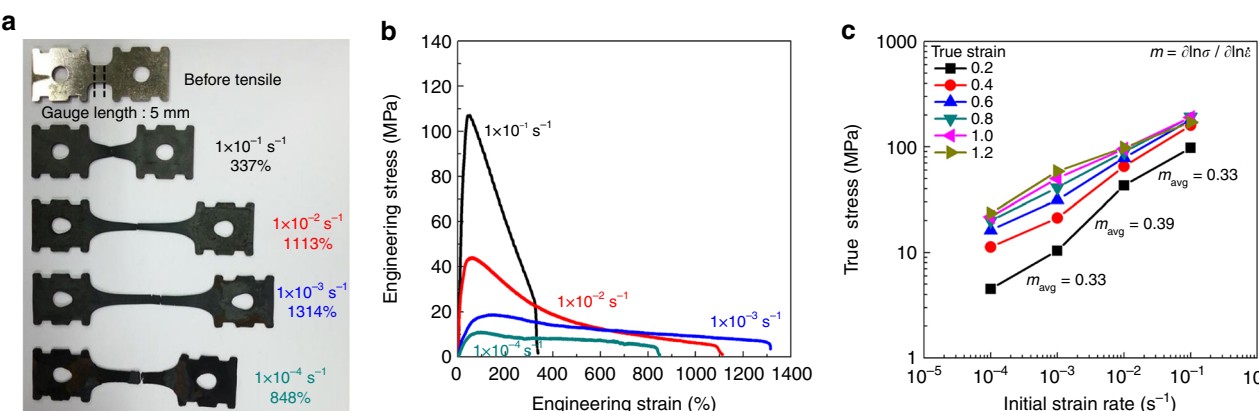

**Fig. 1** Tensile properties of Fe-6.6Mn-2.3Al (wt.%) steel at 850 °C. **a** Images of fractured tensile specimens. **b** Engineering stress–strain curves of tensile specimens strained at various initial strain rates, ranging from $1.0 \times 10^{-4}$ to $1.0 \times 10^{-1}$ s$^{-1}$. **c** Variation of true stress with initial strain rate at specific true strains

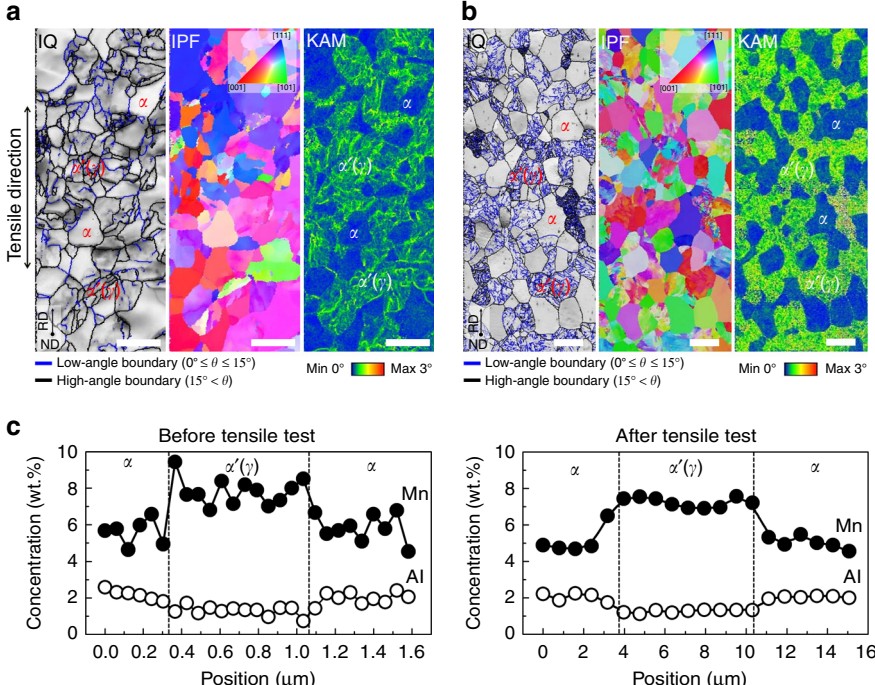

**Fig. 2** SEM-EBSD images of Fe-6.6Mn-2.3Al (wt.%) specimens taken before and after 1314% tensile deformation at 850 °C with a strain rate of $1.0 \times 10^{-3} \, s^{-1}$. EBSD IQ, IPF, and KAM maps of the specimens; **a** Before the tensile test. *Scale bar*, 2 μm. **b** After the tensile test. *Scale bar*, 10 μm. **c** Profiles of Mn and Al concentrations obtained by EDXS line scanning of α ferrite and α′ martensite phases in the specimens before and after the tensile test. EBSD, electron back scatter diffraction; EDXS, energy dispersive X-ray spectroscopy; IPF, inverse pole figure; IQ, image quality; KAM, kernel average misorientation (orientation gradient measure obtained from EBSD)

**Superplasticity mechanism.** To investigate the cause for the outstanding superplasticity of the new steel, the evolutions in shape, chemical composition, crystal orientation, grain size, and volume fraction of each phase were examined by comparing the microstructures of two specimens; one was just held for 5 min at 850 °C without tensile deformation and the other was strained to failure at an initial strain rate of $1.0 \times 10^{-3} \, s^{-1}$ after 5 min holding. As shown in Fig. 2a, b, both α ferrite and α′ martensite, which are formed from γ austenite during cooling from 850 °C to room temperature, were hardly elongated along the tensile direction, i.e., they maintained their equiaxed grain shapes even after superplastic deformation. Grains with high kernel average misorientation (KAM) values are α′ martensite. The chemical composition of each phase also remained practically unaffected by the tensile deformation (Fig. 2c).

The texture evolution observed during forming also points at the occurrence of GBS: while neighboring α grains exhibited similar crystallographic orientations before the tensile test, they revealed random orientations afterwards, as revealed by the electron backscattered diffractometer (EBSD) inverse pole figure (IPF) maps. However, because the EBSD IPF maps were obtained from the limited probing area provided by the tensile specimens, the overall textures of the specimens before and after the tensile test at 850 °C were examined again by X-ray diffraction tests. As shown in Supplementary Fig. 3, whereas the undeformed specimen revealed a strong rotated cube component {001} <110> and a weak γ-fiber <111>//ND (Supplementary Fig. 3a, c), the fractured specimen exhibited a random texture (Supplementary Fig. 3b, d). This confirms the EBSD result that the texture became random during high-temperature tensile deformation. The random texture is a well-known feature associated with boundary sliding and the corresponding random rotations of grains during tensile deformation[11]. Therefore, both, the high $m$ value above 0.3 and the equiaxed grains with random orientations

indicate that superplasticity of the present steel was realized primarily by GBS.

In order to ensure high elongations by superplastic flow, grain size control during high-temperature forming is essential. Here the average grain size of both phases increased from ~1.8 μm (Fig. 2a) to only ~8 μm (Fig. 2b) during tensile deformation at 850 °C for ~4 h. This means that grain growth was so sluggish that the final grain size remained below 10 μm. Above this value, superplastic flow by GBS might otherwise break down. The main reason for the observed slow grain coarsening is the different compositional partitioning of Mn and Al into the ferrite and austenite grains, respectively. This partitioning is known to slow down grain coarsening in duplex alloys because it necessitates substantial diffusional mass transport for the isolated grains of each phase to coarsen[29, 30]. There might be an additional solute drag effect by Mn atoms segregated at the interphase boundaries (Fig. 2c), reducing the mobility of the boundaries. At a low strain rate of $1.0 \times 10^{-4} \, s^{-1}$, the increased grain coarsening due to the prolonged deformation time indeed resulted in reduced elongation (~850%).

Grain coarsening in a dual-phase microstructure is accompanied by the migration of ferrite/ferrite, austenite/austenite, and ferrite/austenite boundaries. The movement of ferrite/austenite boundaries implies the occurrence of phase transformation during the high-temperature tensile test, resulting in the change in phase fraction. Therefore, the ferrite fractions were measured using EBSD images taken at the gauge parts of tensile specimens before and after tensile tests at temperatures of 700–900 °C, probed at initial strain rates of $1.0 \times 10^{-1}$–$1.0 \times 10^{-3} \, s^{-1}$. As shown in Supplementary Fig. 4, the ferrite fraction before the tensile test was higher than the equilibrium ferrite fraction at a given tensile temperature. However, during the tensile test, the ferrite fraction was decreased to almost reach the equilibrium ferrite fraction even at the fast initial strain rate of $1.0 \times 10^{-1} \, s^{-1}$.

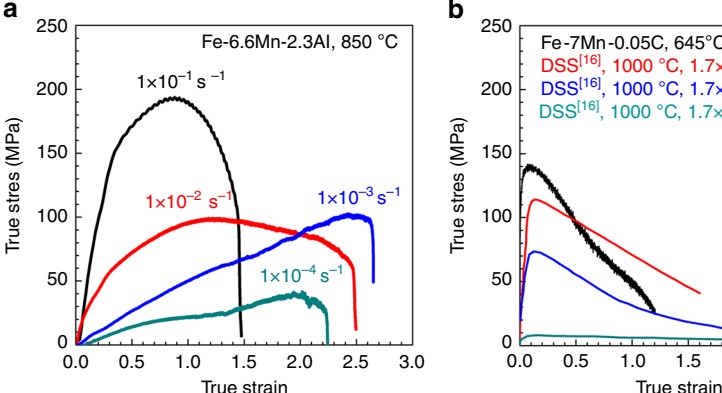

**Fig. 3** True stress–strain curves of medium Mn steels and a duplex stainless steel with dual-phase microstructures of ferrite and austenite at high temperatures. **a** Fe-6.6Mn-2.3Al (wt.%) steel. **b** Fe-7Mn-0.05C (wt.%) steel and a highly alloyed duplex stainless steel (DSS, Fe-25Cr-7Ni-3Mo-0.14N[16])

This confirms the migration of the ferrite/austenite boundaries, i.e., the occurrence of phase transformation, during the tensile test, although the compositional partitioning is relatively slow.

For a more detailed analysis of the superplastic behavior, the engineering stress–strain curves of the present steel (Fig. 1b) were converted to the true stress–strain curves (Fig. 3a). The true stress increased to some extent with increasing true strain, and then decreased again with further strain. The increase in true stress with true strain indicates that strain hardening occurred by dislocation slip inside the grains during tensile deformation. The amount of strain hardening was reduced when the tensile deformation temperature was low and the strain rate was fast (Supplementary Fig. 5). This result is surprising because DSSs and Al-free MMS exhibited little strain hardening when superplastically formed, as shown in Fig. 3b.

Tsuzaki et al.[16] investigated the superplasticity of a DSS with a dual-phase structure of δ ferrite and γ austenite whose grain sizes are ~1 µm. Before the high-temperature tensile tests of cold-rolled specimens, while the ferrite/austenite boundaries had the high misorientation angles, the ferrite/ferrite boundaries had the low misorientation angles. In the early deformation of ~20%, GBS occurred only in the ferrite/austenite boundaries, not in the ferrite/ferrite boundaries. Both ferrite and austenite possessed relatively high dislocation densities due to the difficulty of GBS in the ferrite/ferrite sub-boundaries. With increasing deformation amount, both dislocation slip and dynamic recrystallization occurred primarily in ferrite, not in the austenite with its relatively high solution hardening so that the ferrite/ferrite sub-boundaries changed to the high angle boundaries, leading to active GBS. These results indicate that the DSS with a dual-phase microstructure like the present MMS is deformed primarily by GBS with assistance of dislocation slip. However, strain hardening hardly appeared in the true stress–true strain curves in spite of dislocation slip in ferrite most likely due to the rapid dynamic recovery.

The difference in strain hardening behavior between DSS and MMS results from the difference in the soft phase, where deformation is concentrated. More specific, while ferrite with the rapid dynamic recovery is the soft phase in DSS, resulting in little strain hardening, austenite with the slow dynamic recovery is the soft phase in the present MMS, leading to appreciable strain hardening.

The strain hardening in austenite of the MMS is supported by comparing the strain hardening of C-bearing MMS. Similar to DSS, Fe-7Mn-0.05 C (wt.%) steel with both, C-enriched γ and C-free α phases did not exhibit strain hardening (Fig. 3b) because the C-free soft α grains with the rapid dynamic recovery were first

deformed. However, the present Fe-6.6Mn-2.3Al-0.005C (wt.%) steel with almost C-free γ and α phases revealed strain hardening. This indicates that the strain hardening of the present MMS was most likely due to the deformation of the relatively soft γ phase with its negligible C concentration.

Also, each true stress–true strain curve in Fig. 3a shows a stress peak regardless of strain rate. The peak strain corresponding to the peak stress increases with decreasing strain rate from $1.0 \times 10^{-1}$ to $1.0 \times 10^{-3}\,s^{-1}$, and then decreased again at a strain rate of $1.0 \times 10^{-4}\,s^{-1}$. The decreasing rate of the flow stress after the peak strain was reduced with decreasing strain rate from $1.0 \times 10^{-1}$ to $1.0 \times 10^{-3}\,s^{-1}$, and then increased again at the strain rate of $1.0 \times 10^{-4}\,s^{-1}$. It is well known that the onset of the necking often occurs around the peak strain in the true stress–true strain curves and the reduction in decreasing rate of the flow stress after the peak strain means a higher necking resistance[31, 32]. Therefore, necking is considered to be delayed with decreasing strain rate from $1.0 \times 10^{-1}$ to $1.0 \times 10^{-3}\,s^{-1}$, and then to be accelerated again at the strain rate of $1.0 \times 10^{-4}\,s^{-1}$.

According to Huang et al.[32, 33], necking is closely related to the hardening of the specimen (strain hardening and strain-rate hardening). Although the degree of strain-rate hardening (the m value) was almost constant regardless of strain rate (Fig. 1c), the degree of strain hardening was increased with decreasing strain rate from $1.0 \times 10^{-1}$ to $1.0 \times 10^{-3}\,s^{-1}$, and then reduced again at the strain rate of $1.0 \times 10^{-4}\,s^{-1}$ (Fig. 3a). This indicates that the occurrence of necking was influenced primarily by strain hardening during superplastic deformation. The change in decreasing rate of the flow stress after the peak strain with strain rate is considered to be closely related to the degree of softening mainly by dynamic recrystallization[32].

The differences in both the peak strain and the decreasing rate of the flow stress after the peak strain with strain rate caused the different degrees of tapering of the gauge parts of fractured tensile specimens, i.e., the different degrees of diffuse necking (Fig. 1a).

**Comparison among superplastic alloys.** Figure 4a provides a tensile deformation map compiling the deformation parameters required to realize superplastic elongation above ~300% in various materials. The detailed chemical compositions, thermomechanical pre-treatments, total elongations, and tensile deformation conditions of the superplastic materials are listed in the Supplementary Table 1. The present MMS exhibited superplasticity at the lower temperatures compared to DSS and at the higher strain rates compared to UCS (Fig. 4a). These two advantages are essential for superplastic manufacturing. In

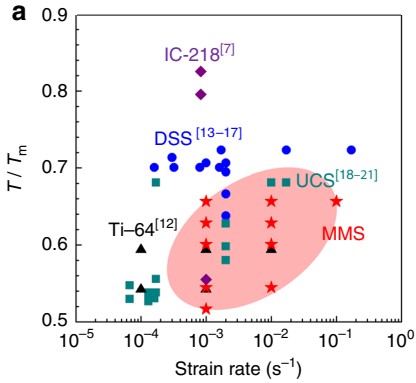
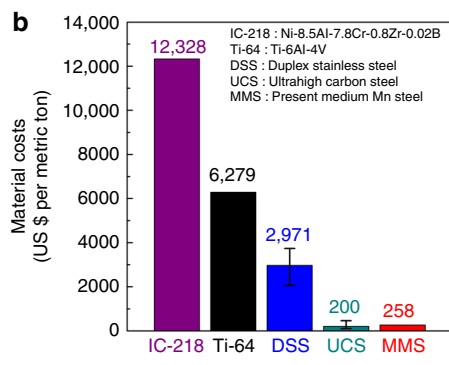

**Fig. 4** Tensile deformation parameter maps and material costs of various superplastic alloys including the present Fe-6.6Mn-2.3Al (wt.%) steel. **a** Tensile deformation temperature over melting temperature ($T/T_m$)–strain rate ($s^{-1}$) maps for superplasticity. **b** Comparison of material costs (US $ per metric ton) of superplastic alloys

addition, the MMS is fabricated by simple conventional hot- and cold-rolling process, unlike the complicated thermomechanical pre-treatment required for the UCS. Figure 4b compares the material costs of several superplastic materials. They were calculated based on the average prices of alloying elements over the past 4 years (November 2012–November 2016) (http://www.kores.net). The costs of the MMS are much below those of the Ni alloy (IC-218), Ti alloy (Ti-64), and even DSS. Although the material costs of a UCS are slightly below those of the present MMS, the complicated thermomechanical pre-treatment of UCS deteriorates productivity, resulting in a high price of the final product. Also, such high C-containing steels cannot be welded, an essential point in downstream manufacturing.

Therefore, the novel MMS is a new, plain, superplastic, ferrous material with a combination of low-deformation temperature, high strain rate, simple fabrication, and several economic advantages. These features render MMSs highly suited for superplastic mass production of complex parts. Also, since MMSs exhibit high room-temperature tensile strength above 700 MPa after superplastic deformation (Supplementary Fig. 6), it is a most promising material for mechanically heavily stressed parts of vehicles requiring both complicated shapes and high strength.

## Methods

**Sample preparation**. A 30 kg ingot of a newly designed medium Mn steel with the chemical composition of Fe-6.6Mn-0.005C-2.3Al (wt.%) was cast using a vacuum induction furnace. After homogenization at 1200 °C for 12 h, the ingot was hot-rolled to a 7.0-mm-thick plate at ~1100 °C, followed by water quenched to room temperature to obtain an almost full $\alpha'$ martensitic microstructure (Supplementary Fig. 7). After surface descaling, the hot-rolled plate was cold-rolled to a 1.5-mm-thick sheet with the thickness reduction of 80%. The equilibrium phase diagram of the present MMS was calculated using Thermo-Calc software with TCFE7 database[34].

**Microstructural observation**. Microstructures were observed using an optical microscope (OM; Olympus, BX41M), a field-emission scanning electron microscope (FE-SEM; JEOL, JSM-7001F) equipped with an EBSD (EDAX-TSL, Digiview), a field-emission transmission electron microscope (FE-TEM; JEOL, JEM-2100F), and an X-ray diffractometer (XRD; Bruker, D8 Advance). The accelerating voltage, probe current, working distance, and step size for the operation of the SEM were 20 kV, 12 nA, 15 mm, and 35 nm, respectively. The chemical composition of each phase was measured by energy dispersive X-ray spectroscopes (EDXS) attached to a transmission electron microscope (TEM) for the undeformed specimen and to SEM for the tensile-strained specimen. The SEM and EBSD specimens were mechanically polished using a suspension including 0.04 μm colloidal silica particles, and then electro-polished in a mixed solution of 90% glacial acetic acid (CH3COOH) and 10% perchloric acid (HClO4) at 15 V for 1 min to remove the damaged layer. The EBSD KAM analysis was conducted using an average misorientation angle around a measurement point with respect to a defined set of the nearest neighbor points. Thin foils for TEM observation were

jet-polished in the same mixed solution used for the EBSD sample. The overall textures of specimens before and after the tensile test at 850 °C were investigated by measuring three pole figures of {110}, {200}, and {211} of α ferrite and α′ martensite phases using the XRD with Co-K$_\alpha$ radiation ($\lambda = 0.1789$ nm). The orientation distribution functions were converted from the pole figure data.

**Tensile test**. Tensile specimens were machined into a dog-bone shape along the cold-rolling direction. The size of the gauge portion was 1.5 mm in thickness, 10.0 mm in width, and 5.0 mm in length. Before starting the tensile tests each sample was first heated to the corresponding deformation temperature with a heating rate of 0.5 °C s$^{-1}$ and then held for 5 min at the corresponding deformation temperature for the homogenization of specimen temperature and austenite formation. Then the tensile tests were conducted at initial strain rates ranged from $1 \times 10^{-1}$ to $1 \times 10^{-4}$ s$^{-1}$ at elevated temperatures using an MTDI universal testing machine. After tensile failure, the specimens were removed from the furnace and air-cooled to room temperature. The data of true stress and true strain were converted from the measured data of load and displacement by assuming volume constancy and homogeneous deformation[27, 32, 33].

**Data availability**. The data supporting the findings of this study are available from the corresponding author on request.

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

## Acknowledgements

This research was supported by the Basic Science Research Program through the National Research Foundation of Korea (NRF) funded by the Ministry of Education (grant number: NRF-2015R1D1A1A09059936) and by the third Stage of Brain Korea 21 Plus Project in 2017. We are thankful to Dong-Hyun Lee and Professor Jae-il Jang (Hanyang University, Seoul) for their help with nano-indentation tests, and to Sang-Chul Kwon and Professor Hyo-Tae Jeong (Gangneung-wonju National University, Gangneung) for their assistance in X-ray pole figure measurements.

## Author contributions

J.H. designed alloy composition, performed major experiments, analyzed the results, and wrote the first version of manuscript. M.K. revised the manuscript. S.-H.K. performed major experiments and drew the figures. S.-J.L. performed hot and cold rolling, and prepared the specimens for microstructure observation. H.-J.L. performed the high-temperature tensile tests. Y.-K.L. first came up with the idea of superplasticity of medium Mn alloy, supervised the present research, designed the alloy composition, analyzed the results, and significantly revised the manuscript. D.P. and D.R. revised the text and contributed to the discussion. All authors discussed the results and commented on the manuscript.

## Additional information

**Competing interests:** The authors declare no competing financial interests.

**Change history:** A correction to this article has been published and is linked from the HTML version of this paper.

