## [Peer Review File · Nature Communications]

Reviewers' comments:

Reviewer #1 (Remarks to the Author):

The results that are reported in this paper are certainly novel and interesting for a metallurgy audience. The question is about the significance of the huge superplasticity of 1300% that is reported. To what extent is this really extraordinary ? or surprising ? and can this measurement be compared to other literature data ? This is probably the main point of criticism (that is explained in more detail hereafter) to decide whether this paper can be considered as interesting for a wider field (in terms of a breakthrough result or not), but there are other questions and comments:

- Based on the information reported in the paper, it is extremely difficult to determine if the 1300% ductility is really remarkable. The map of Fig. 4a is certainly useful to compare materials but, the information on the ductility is absent (this is only for materials showing >300% ductility).
- Regarding the experimental measurements, the gauge length of the sample looks really short compared to standard conditions for uniaxial tension. The short length will certainly affect the conditions for necking compared to long specimens. Short specimen resist much better to plastic localization than long specimens. So the question is to what extent the 1300% is not an artificial number compared to other studies using long tensile test samples ?
- Regarding plastic localization, the authors could have also consulted the series of papers by Hutchinson and Neale on the topic – which addresses the effect of m and other factors in a more general way than only under the scope of superplasticity.
- In the first sentence of the text, the authors insist on TRIP steels. Is the steel addressed in this study exhibiting a TRIP effect ? Does it show a TRIP effect after superplastic forming ? If not, then the reference to TRIP steels at the beginning of the paper is misleading, otherwise this point should be clarified with data after superplastic forming.
- Following the last comment, data showing the behavior of the steel after superplastic deformation are missing – only the tensile strength above 700MPa is mentioned.
- Regarding the deformation mechanisms, the analysis provided is certainly done with state of the art methods, but lacks a critical comparison with the mechanisms usually reported in the literature on superplasticity in similar dual phase structures.
- In Fig. 3, the true stress / true strain curves are reported without explaining the methodology to extract true stress and true strain from these specimens for which the cross-section is not constant as observed in Fig. 1 and 2.
- In general, there is a lack of supplementary information on the methods and on additional supporting data.

Based on these comments, it is difficult to determine at this stage if the paper contains de degree of novelty and impact which justifies a publication in Nature Communication.

Reviewer #2 (Remarks to the Author):

Medium Mn TRIP steels are currently being subjected to extensive research in academia and industry as they are potential materials for lightweight automotive applications. Most existing research on medium Mn TRIP steels focus on their mechanical properties at room temperature. Nevertheless, the present manuscript focuses on a new and interesting point, i.e., superplasticity at high temperature, which opens up a new potential application for medium TRIP steels. The superplasticity property of the reported medium Mn TRIP steel is excellent (1300%) and suitable for industrial applications. Furthermore, the steel has a lean chemical composition so that it is easy for mass production and has a low cost. In a word, this manuscript presents a new and interesting aspect of the medium Mn steel

and is worth to be published in Nature Communications. Nevertheless, the authors are advised to consider the following minor points before the paper is accepted.

1. Please address how the true stress-strain curve is obtained (Fig. 3a). In order to obtain a correct true stress-strain curve, one should take care the necking during the tensile tests. The authors claimed that there is no necking during tensile tests. But the engineering stress-strain curves (Fig. 1b) shows that there is a peak stress at the engineering stress-strain curve, which may corresponds to the necking. The authors are advised to address this point further.
2. It claims that there is strain hardening during the tensile tests as shown in Fig. 3a. Nevertheless, if the true stress-strain curves are not properly obtained, one cannot claim the strain hardening based on these true stress-strain curves. If the strain hardening indeed took place by dislocations during tensile tests, one may expected that the dislocation density in the ferrite or austenite should be higher. The reviewer recommends that the authors check the nanohardness of ferrite after different interrupted tensile tests. If the hardness is indeed increased, the strain hardening is verified. It is difficult to measure nanohardness of the austenite for such interrupted tensile tests, as they will transform martensite after cooled to room temperature. Furthermore, some nanohardness tests should be also carried out on the samples which was hold at the same temperature (say 850 oC) for the same duration without being subjected to any deformation. In other words, it is recommended to perform extra nanohardness tests on samples subjected to different interrupted tensile strains at 850 oC to verify whether or not strain hardening indeed took place during the tensile test.

Reviewer #3 (Remarks to the Author):

The manuscript introduces a new class of superplastic alloy steel exhibiting superior ductility at a temperature as low as 650 °C. This unique property is attributed to concurrent grain boundary sliding and dislocation hardening mechanisms. The designed alloy (i.e., Fe-6.6Mn-2.3Al-0.005) is relatively cheap compared with the conventional superplastic alloys and is produced through a simple thermomechanical processing route. This is a well-written and well-presented manuscript. This paper can gather much interest in the society of materials scientists. I recommend the publication of manuscript provided that the following points have been addressed.

- 1) What was the initial microstructure before reheating to the tensile testing temperature (e.g., 850°C)? Please include the initial microstructure.
- 2) The grain growth would be expected to take place during reheating to the tensile testing temperature. Please comment on the grain size change at different reheating temperatures (i.e. 650°C to 900°C) and its contribution to the superplasticity of the alloy.
- 3) The authors claimed that the texture became random because of the grain boundary sliding, which was concluded from an IPF image of a limited area. To prove the claim, they need to provide the overall texture of sample before and after the tensile testing.
- 4) The grain growth was observed during the tensile testing, which was affected by the tensile temperature. As the grain growth results from the movement of ferrite/ferrite and ferrite/austenite interfaces, the phase transformation would be, to some extent, expected during the tensile testing, though the compositional partitioning is relatively slow. Please provide the volume fraction of phases at different tensile conditions, as the extent of phase change depends on the tensile condition, namely deformation temperature, strain rate and strain.
- 5) Please provide the strain at which the tensile testing was ceased in Figure 2 caption.

Reviewer #1

Overall comments) The results that are reported in this paper are certainly novel and interesting for a metallurgy audience. The question is about the significance of the huge superplasticity of 1300% that is reported. To what extent is this really extraordinary? or surprising ? and can this measurement be compared to other literature data? This is probably the main point of criticism (that is explained in more detail hereafter) to decide whether this paper can be considered as interesting for a wider field (in terms of a breakthrough result or not), but there are other questions and comments:

Q1) Based on the information reported in the paper, it is extremely difficult to determine if the 1300 % ductility is really remarkable. The map of Fig. 4a is certainly useful to compare materials but, the information on the ductility is absent (this is only for materials showing >300 % ductility).

A1) Detailed information including **ductility** (e.g. alloy composition, thermomechanical process, total elongation, deformation temperature and strain rate) of the present medium Mn steel and other superplastic alloys is already listed in the Supplementary Table 1 of the present manuscript. The ductilities of Ni alloy, Ti alloy, duplex stainless steel, and ultrahigh C steel are 302 % - 638 % ⁷, 380 % - 850 % ¹², 574 % - 2500 % ¹³⁻¹⁷, and 380 % - 1300 % ¹⁸⁻²¹, respectively. The ductility of the present medium Mn steel ranges from 306 % to 1314 %. Accordingly, it is difficult to tell that the 1300 % ductility of the present medium Mn steel is really extraordinary. **Therefore, the novelty of the present paper is not “1300 % ductility”.** **As we mentioned in the last part of Introductory paragraph, the originality of our work is to find out a new compositionally lean, superplastic medium Mn steel with low**

material cost, simple fabrication process and low superplastic temperature. Until now, only duplex stainless steel and ultrahigh C steel are known as superplastic ferrous alloys. These alloys have the limitations to commercialization due to high material cost, complex fabrication process, and high deformation temperature. However, here we suggest a new superplastic steel which resolves these limitations.

In addition, in a practical point of view, ductility over 300 % is good enough to make mechanical parts with complex shapes. Thus, we compared superplastic materials with ductility above 300 % in Fig. 4.

Q2) Regarding the experimental measurements, the gauge length of the sample looks really short compared to standard conditions for uniaxial tension. The short length will certainly affect the conditions for necking compared to long specimens. Short specimen resist much better to plastic localization than long specimens. So the question is to what extent the 1300% is not an artificial number compared to other studies using long tensile test samples? Regarding plastic localization, the authors could have also consulted the series of papers by Hutchinson and Neale on the topic – which addresses the effect of m and other factors in a more general way than only under the scope of superplasticity.

A2) We found an article reporting the relationship between gauge length and total elongation of superplastic alloy (P. S. Bate *et al.*, Mater. Sci. Technol. **24** (2008) 1265-1270). In this article, authors made the tensile specimens of Al-4.5Mg alloy with different gauge lengths (6.3 mm, 12.7 mm, and 25.4 mm) and compared their ductility at 530 °C. The measured ductility was ~400 % for the specimen with the gauge length of 6.3 mm, ~300 % for the specimen with the gauge length of 12.7 mm, and ~200 % for the specimen with the gauge length of 25.4 mm. Bate *et al.* reported that a tensile specimen with a shorter gauge length has

the higher ductility due to the drawing-in of material from the specimen ends to the gauge part. Therefore, we agree that 1300 % ductility may be reduced to some extent for tensile specimens with the longer gauge length than the gauge length of 5 mm used in the present study. However, as we mentioned in A1, we do not think that “1300 % ductility” itself is our novelty.

In addition, as Bate *et al.* mentioned, there are no internationally agreed specifications for testpiece geometry for superplastic materials. Actually, the gauge lengths of tensile specimens used in the references of the present manuscript are as follows.

Ref.	1st author	Gauge length (mm)
4	Mohamed, F. A.	6.35
6	Kawasaki, M.	1
8	McFadden, S. X.	1
13	Smith, C. I.	10
14	Maehara, Y	8
15	Maehara, Y	8
16	Tsuzaki, K.	10
17	Sagradi, M.	5.2
19	Wadsworth, J.	5.1
20	Ruano, O. A.	5.08
21	Zhang, H.	10
22	Zhang, H.	10

As can be seen in this table, the gauge length of tensile specimens is shorter than ~10 mm. Some researchers used the similar gauge length (~5 mm) [Refs 17, 19 and 20] and even

shorter gauge length (1 mm), compared to the gauge length of 5 mm used in this study [Refs 6 and 8]. Therefore, we do not think that the gauge length (5 mm) of tensile specimens used in the present study gave rise to the wrong ductility values.

Q3) In the first sentence of the text, the authors insist on TRIP steels. Is the steel addressed in this study exhibiting a TRIP effect? Does it show a TRIP effect after superplastic forming? If not, then the reference to TRIP steels at the beginning of the paper is misleading, otherwise this point should be clarified with data after superplastic forming.

A3) I agree with the reviewer's comment that the reference to TRIP steels at the beginning of the paper can mislead the readers. Superplasticity is not caused by the TRIP effect and TRIP does not occur at room temperature after superplastic deformation due to the absence of retained austenite. Therefore, to avoid misleading we simply deleted the word TRIP in the line 19 on the page 2 of the Introduction as follows:

“Fe-medium Mn ~~transformation induced plasticity (TRIP)~~ steels”

Q4) Following the last comment, data showing the behavior of the steel after superplastic deformation are missing – only the tensile strength above 700 MPa is mentioned.

A4) We performed the room-temperature tensile tests at an initial strain rate of $1.0 \times 10^{-3} \text{ s}^{-1}$ using the specimens annealed at 850 °C. The annealing time was 5 min, 25 min, and 224 min which correspond to the times elapsed until tensile failure at 850 °C with the strain rates of $1.0 \times 10^{-1} \text{ s}^{-1}$, $1.0 \times 10^{-2} \text{ s}^{-1}$ and $1.0 \times 10^{-3} \text{ s}^{-1}$, respectively.

As we mentioned in the manuscript, the high yield strength and tensile strength above ~700 MPa were obtained most likely due to both the fine grain size and the hard

martensite phase. According to the reviewer's comment, the following figures were added to the revised manuscript as Supplementary Figure 6.

Supplementary Figure 6. Variation of room-temperature tensile properties with annealing time at 850 °C in Fe-6.6Mn-2.3Al (wt. %) steel. (a) Engineering stress-strain curves measured at the initial strain rate of $1.0 \times 10^{-3} \text{ s}^{-1}$. **(b)** Yield strength, ultimate tensile strength, and elongation.

Q5) Regarding the deformation mechanisms, the analysis provided is certainly done with state of the art methods, but lacks a critical comparison with the mechanisms usually reported in the literature on superplasticity in similar dual phase structures.

A5) According to the reviewer's comment, we read literature reporting on the superplasticity of duplex stainless steel with a dual-phase structure of δ -ferrite and γ -austenite like the present medium Mn steel and compared high-temperature deformation mechanisms between the two steels in line 2 on page 7 in the Result section of the revised manuscript.

Revised: Tsuzaki *et al.* investigated the superplasticity of a duplex stainless steel with a dual-phase structure of δ ferrite and γ austenite whose grain sizes are $\sim 1 \mu\text{m}$ [Ref. 16]. Before the

high-temperature tensile tests of cold-rolled specimens, whereas the ferrite/austenite boundaries had the high misorientation angles, the ferrite/ferrite boundaries had the low misorientation angles. In the early deformation of ~20 %, GBS occurred only in the ferrite/austenite boundaries, not in the ferrite/ferrite boundaries. Both ferrite and austenite possessed relatively high dislocation densities due to the difficulty of GBS in the ferrite/ferrite sub-boundaries. With increasing deformation amount, both dislocation slip and dynamic recrystallization occurred primarily in ferrite, not in austenite with relatively high solution hardening so that the ferrite/ferrite sub-boundaries changed to the high angle boundaries, leading to active GBS. These results indicate that the duplex stainless steel with a dual-phase microstructure like the present medium Mn steel is deformed primarily by GBS with assistance of dislocation slip. However, strain hardening hardly appeared in the true stress-true strain curves in spite of dislocation slip in ferrite most likely due to the rapid dynamic recovery.

The difference in strain hardening behavior between duplex stainless steel and medium Mn steel results from the difference in the soft phase, where deformation is concentrated. Namely, while ferrite with the rapid dynamic recovery is the soft phase in duplex stainless steel, resulting in little strain hardening, austenite with the slow dynamic recovery is the soft phase in the present medium Mn steel, leading to appreciable strain hardening.

Q6) In Fig. 3, the true stress/true strain curves are reported without explaining the methodology to extract true stress and true strain from these specimens for which the cross-section is not constant as observed in Fig. 1 and 2.

A6) In our study, true stress-true strain curves were converted from the load-displacement

data by assuming volume constancy and homogeneous deformation according to the following papers.

- Figueiredo, R. B. and Langdon, T.G., *J. Mater. Res. Technol.* **6** (2017) 129 [Ref. 27]
- Huang, L. *et al.*, *Mater. Sci. Eng. A* **647** (2015) 277 [Ref. 31]
- Huang, L. *et al.*, *Mater. Sci. Eng. A* **634** (2015) 71 [Ref. 32]

Therefore, we added the methodology to extract true stress and true strain to the Methods part (at line 17 on page 11).

Revised: The data of true stress and true strain were converted from the measured data of load and displacement by assuming volume constancy and homogeneous deformation [Refs 27, 31 and 32].

Q7) In general, there is a lack of supplementary information on the methods and on additional supporting data.

A7) The original manuscript had three supplementary figures and one supplementary table. But, in the revised manuscript four supplementary figures answering the reviewers' comments were additionally annexed as follows.

1. Change in overall texture of Fe-6.6Mn-2.3Al (wt. %) steel before and after the tensile test at 850 °C at an initial strain rate of $1.0 \times 10^{-3} \text{ s}^{-1}$. (Supplementary Figure 3)
2. Variation of ferrite fraction with initial strain rate and tensile deformation temperature in Fe-6.6Mn-2.3Al (wt. %) steel. (Supplementary Figure 4)
3. Variation of room-temperature tensile properties with annealing time at 850 °C in Fe-6.6Mn-2.3Al (wt. %) steel. (Supplementary Figure 6)
4. Optical microstructures of Fe-6.6Mn-2.3Al (wt. %) steel. (Supplementary Figure 7)

Reviewer #2

Q1) Please address how the true stress-strain curve is obtained (Fig. 3a). In order to obtain a correct true stress-strain curve, one should take care the necking during the tensile tests. The authors claimed that there is no necking during tensile tests. But the engineering stress-strain curves (Fig. 1b) show that there is a peak stress at the engineering stress-strain curve, which may correspond to the necking. The authors are advised to address this point further.

A1) In our study, true stress-true strain curves were converted from the load-displacement data by assuming volume constancy and homogeneous deformation according to the following papers.

- Figueiredo, R. B. and Langdon, T.G., *J. Mater. Res. Technol.* **6** (2017) 129 [Ref. 27]
- Huang, L. *et al.*, *Mater. Sci. Eng. A* **647** (2015) 277 [Ref. 31]
- Huang, L. *et al.*, *Mater. Sci. Eng. A* **634** (2015) 71 [Ref. 32]

Therefore, we added the methodology to extract true stress and true strain to the Methods part (at line 17 on page 11).

Revised: The data of true stress and true strain were converted from the measured data of load and displacement by assuming volume constancy and homogeneous deformation [Refs 27, 31 and 32].

Meanwhile, we also read some literature reporting on the necking behavior and the stress peak in the true stress-true strain curve and added the analysis of their relationship in

the revised manuscript as follows.

Original: strain localization and necking are suppressed during plastic deformation

Revised: strain localization and localized necking (but not diffuse necking) are suppressed during plastic deformation (line 5 page 4)

Also, the following paragraphs are added in line 6 on page 8.

"Meanwhile, each true stress-true strain curve in Fig. 3a shows a stress peak regardless of strain rate. The peak strain corresponding to the peak stress increases with decreasing strain rate from $1.0 \times 10^{-1} \text{ s}^{-1}$ to $1.0 \times 10^{-3} \text{ s}^{-1}$, and then decreased again at a strain rate of $1.0 \times 10^{-4} \text{ s}^{-1}$. The decreasing rate of the flow stress after the peak strain was reduced with decreasing strain rate from $1.0 \times 10^{-1} \text{ s}^{-1}$ to $1.0 \times 10^{-3} \text{ s}^{-1}$, and then increased again at the strain rate of $1.0 \times 10^{-4} \text{ s}^{-1}$. It is well known that the onset of the necking often occurs around the peak strain in the true stress-true strain curves and the reduction in decreasing rate of the flow stress after the peak strain means a higher necking resistance^{30,31}. Therefore, necking is considered to be delayed with decreasing strain rate from $1.0 \times 10^{-1} \text{ s}^{-1}$ to $1.0 \times 10^{-3} \text{ s}^{-1}$, and then to be accelerated again at the strain rate of $1.0 \times 10^{-4} \text{ s}^{-1}$.

According to Huang *et al.* [Refs 31 and 32], necking is closely related to the hardening of the specimen (strain hardening and strain-rate hardening). Whereas the degree of strain-rate hardening (the m value) was almost constant regardless of strain rate (Fig. 1c), the degree of strain hardening was increased with decreasing strain rate from $1.0 \times 10^{-1} \text{ s}^{-1}$ to $1.0 \times 10^{-3} \text{ s}^{-1}$, and then reduced again at the strain rate of $1.0 \times 10^{-4} \text{ s}^{-1}$ (Fig. 3a). This indicates that the occurrence of necking was influenced primarily by strain hardening during superplastic deformation. The change in decreasing rate of the flow stress after the peak strain with strain rate is considered to be closely related to the degree of softening mainly by

dynamic recrystallization [Ref. 31].

The differences in both the peak strain and the decreasing rate of the flow stress after the peak strain with strain rate caused the different degrees of tapering of the gauge parts of fractured tensile specimens, i.e. the different degrees of diffuse necking (Fig. 1a). "

Q2) It claims that there is strain hardening during the tensile tests as shown in Fig. 3a. Nevertheless, if the true stress-strain curves are not properly obtained, one cannot claim the strain hardening based on these true stress-strain curves. If the strain hardening indeed took place by dislocations during tensile tests, one may expect that the dislocation density in the ferrite or austenite should be higher. The reviewer recommends that the authors check the nanohardness of ferrite after different interrupted tensile tests. If the hardness is indeed increased, the strain hardening is verified. It is difficult to measure nanohardness of the austenite for such interrupted tensile tests, as they will transform to martensite after cooled to room temperature. Furthermore, some nanohardness tests should be also carried out on the samples which were held at the same temperature (say 850 °C) for the same duration without being subjected to any deformation. In other words, it is recommended to perform extra nanohardness tests on samples subjected to different interrupted tensile strains at 850 °C to verify whether or not strain hardening indeed took place during the tensile test.

A2) As the reviewer suggested, we also thought that nano-indentation would be useful to examine which phase was strain-hardened during high-temperature tensile tests even before the submission of our manuscript. Thus, we carried out nano-indentation tests using three different types of samples; a undeformed specimen held for 5 min at 850 °C before the tensile test, a gauge part of tensile specimen fractured at a strain rate of $1.0 \times 10^{-3} \text{ s}^{-1}$ at 850 °C, and a specimen held at 850 °C for the same tensile duration without any tensile deformation. The

reason that the fractured specimen was used instead of specimens interrupted during tensile deformation is that strain hardening continued to occur until failure so that the fractured specimen exhibited the highest strain hardening, as shown in Fig. 3a. Each type of samples was indentation tested three times using three different specimens. Namely, total 9 specimens underwent nano-indentation tests at room temperature after careful surface polishing. The indentation was carried out 150 times on the surface of each specimen using a Nanoindenter-XP (MTS Corp., Oak Ridge, TN) with a typical three-sided pyramidal Berkovich indenter tip. Because the grain sizes ($\sim 2\text{-}8\ \mu\text{m}$) of ferrite and martensite were small, a small load of 5 mN was employed. After indentation, only the hardness data measured within the grains were collected to avoid the grain boundary effect. As a result, as shown in the below figure, the range of the hardness values of ferrite in three different types of samples was overlapped so that it was difficult to observe strain hardening in ferrite. The similar result was also obtained in the case of martensite. Therefore, we think that it is very difficult to figure out strain hardening behavior by means of nano-indentation.

In addition, as we already mentioned in the original manuscript, we think that austenite, not ferrite, was strain-hardened during high-temperature tensile deformation. However, unfortunately austenite changed to martensite during cooling to room temperature

after tensile tests. Thus, it is difficult to confirm the strain hardening of austenite through nano-indentation testing, as the reviewer also understands already.

Reviewer #3

Q1) What was the initial microstructure before reheating to the tensile testing temperature (e.g., 850 °C)? Please include the initial microstructure.

A1) The microstructures of both hot- and cold-rolled specimens before reheating to tensile testing temperatures were full α' martensite, as shown in the following figure. We added this figure as Supplementary Figure 7 in the revised manuscript, and mentioned it in the Methods part in line 9 on page 10.

Revised:

Supplementary Figure 7. Optical microstructures of Fe-6.6Mn-2.3Al (wt. %) steel. (a) Hot-rolled specimen. **(b)** Cold-rolled specimen.

Q2) The grain growth would be expected to take place during reheating to the tensile testing

temperature. Please comment on the grain size change at different reheating temperatures (i.e. 650 °C to 900 °C) and its contribution to the superplasticity of the alloy.

A2) According to the reviewer's comment, the grain sizes of the specimens, which were held for 5 min before tensile tests at temperatures ranging from 650 °C to 900 °C, were measured by a linear intercept method using SEM images and EBSD image quality (IQ) maps. As expected, the grain size was increased from 0.8 μm to 2.2 μm with increasing tensile temperature from 650 °C to 900 °C, as shown in the below figure. Accordingly, in the viewpoint of grain size, it is expected that the specimen strained at a higher tensile temperature exhibits a lower superplasticity. However, the increase in grain size with tensile temperature is not significant. In addition, phase fraction as well as grain size changes according to tensile temperature and contributes to the superplasticity. Because tensile temperature itself also influences grain boundary sliding, it is difficult to separate the contribution of grain size from the contributions of phase fraction and tensile temperature.

Q3) The authors claimed that the texture became random because of the grain boundary sliding, which was concluded from an IPF image of a limited area. To prove the claim, they need to provide the overall texture of sample before and after the tensile testing.

A3) According to the reviewer's comment, we measured the overall textures at the gauge parts of two specimens before and after the tensile test; a tensile specimen was held for 5 min at 850 °C without tensile deformation and the other specimen was strained to failure at an initial strain rate of $1.0 \times 10^{-3} \text{ s}^{-1}$ after a holding of 5 min. The overall textures of specimens before and after the tensile test at 850 °C were investigated by measuring three pole figures of {110}, {200} and {211} using an X-ray diffractometer with Co- K_{α} radiation ($\lambda=0.1789$ nm). The orientation distribution functions (ODF) were also converted from the pole figure data. As shown in Supplementary Fig. 3, whereas the undeformed specimen revealed a strong rotated cube component {001}<110> and a weak γ -fiber <111>//ND, the fractured specimen exhibited a random texture. This confirmed the EBSD result that the texture became random during high-temperature tensile deformation.

Therefore, we added a figure showing the overall textures before and after the tensile test at 850 °C as Supplementary Figure 3 in the revised manuscript and explained the figure in line 6 on page 5 in the Result and some parts in the Method as follows.

Revised:

(Result) However, because the EBSD IPF maps were obtained from the limited area of tensile specimens, the overall textures of the specimens before and after the tensile test at 850 °C were examined again by the X-ray diffraction tests. As shown in Supplementary Fig. 3, whereas the undeformed specimen revealed a strong rotated cube component {001}<110> and a weak γ -fiber <111>//ND (Supplementary Figs. 3a,c), the fractured specimen exhibited a

random texture (Supplementary Figs. 3b,d). This confirms the EBSD result that the texture became random during high-temperature tensile deformation.

(Method)

Page 10, line 17: and an X-ray diffractometer (XRD; Bruker, D8 Advance)

Page 11, line 3: The overall textures of specimens before and after the tensile test at 850 °C were investigated by measuring three pole figures of {110}, {200}, and {211} of α ferrite and α' martensite phases using the XRD with Co- K_{α} radiation ($\lambda=0.1789$ nm). The orientation distribution functions (ODF) were converted from the pole figure data.

Supplementary Figure 3. Change in overall texture of Fe-6.6Mn-2.3Al (wt. %) steel before and after the tensile test at 850 °C at an initial strain rate of $1.0 \times 10^{-3} \text{ s}^{-1}$. (a) Pole

figure before the tensile test. **(b)** Pole figure after the tensile test. **(c)** Orientation distribution function before the tensile test. **(d)** Orientation distribution function after the tensile test.

Q4) The grain growth was observed during the tensile testing, which was affected by the tensile temperature. As the grain growth results from the movement of ferrite/ferrite and ferrite/austenite interfaces, the phase transformation would be, to some extent, expected during the tensile testing, though the compositional partitioning is relatively slow. Please provide the volume fraction of phases at different tensile conditions, as the extent of phase change depends on the tensile condition, namely deformation temperature, strain rate and strain.

A4) According to the reviewer's comment, we measured the change in ferrite fraction of the specimens with tensile testing conditions (i.e. temperature: 700 °C to 900 °C, strain rate: $1.0 \times 10^{-1} \text{ s}^{-1}$ to $1.0 \times 10^{-3} \text{ s}^{-1}$). The ferrite fraction was measured at the gauge part of tensile specimens before and after tensile tests using the SEM and the EBSD. As shown in the below figure, ferrite fraction before the tensile tests was higher than equilibrium ferrite fraction at a given temperature. However, during tensile deformation ferrite fraction was decreased to almost reach equilibrium ferrite fraction even at a fast strain rate of $1.0 \times 10^{-1} \text{ s}^{-1}$. This means that the ferrite/austenite boundaries migrated so that phase transformation occurred during the tensile test, as the reviewer expected.

Accordingly, we added a figure showing the change in ferrite fraction as functions of initial strain rate and tensile temperature as Supplementary Figure 4 in the revised manuscript and explain the figure in line 6 on page 6 in the Result as follows.

Revised: Grain coarsening in a dual-phase microstructure is accompanied by the migration of

ferrite/ferrite, austenite/austenite, and ferrite/austenite boundaries. The movement of ferrite/austenite boundaries implies the occurrence of phase transformation during the high-temperature tensile test, resulting in the change in phase fraction. Therefore, the ferrite fractions were measured using EBSD images taken at the gauge parts of tensile specimens before and after tensile tests at temperatures of 700 °C - 900 °C with the initial strain rates of $1.0 \times 10^{-1} \text{ s}^{-1}$ - $1.0 \times 10^{-3} \text{ s}^{-1}$. As shown in Supplementary Fig. 4, the ferrite fraction before the tensile test was higher than the equilibrium ferrite fraction at a given tensile temperature. However, during the tensile test, the ferrite fraction was decreased to almost reach the equilibrium ferrite fraction even at the fast initial strain rate of $1.0 \times 10^{-1} \text{ s}^{-1}$. This confirms the migration of the ferrite/austenite boundaries, i.e. the occurrence of phase transformation, during the tensile test, although the compositional partitioning is relatively slow.

Supplementary Figure 4. Variation of ferrite fraction with initial strain rate and tensile deformation temperature in Fe-6.6Mn-2.3Al (wt. %) steel.

Q5) Please provide the strain at which the tensile testing was ceased in Figure 2 caption.

A5) The strain was 1314 %; this value was added into the caption of Figure 2 in the revised manuscript.

Thank you very much for your kind review.

Sincerely yours,

Prof. Young-Kook Lee

E-mail: yklee@yonsei.ac.kr

REVIEWERS' COMMENTS:

Reviewer #1 (Remarks to the Author):

The revision has allowed the authors to clarify several important points. They have convincingly answered to my remarks, except perhaps for a few points which require minor revision.

1. Regarding the “breakthrough–originality” of the work. My point was to support the fact that reaching 1300% ductility is a breakthrough. The authors answer that the key element is to have a compositionally lean superplastic material more than 1300% ductility. Fine, I’m ready to follow this argument, certainly, and to recognize that the achievement is very significant under this view. But then, the title should be rephrased – because the title makes the reader believe that the focal point is the super high ductility. Perhaps just add the word “lean” in the title “xxx in a lean Fe-Mn alloy” or something like that.

2. Regarding the question of the gauge length which has been selected to perform the tests, the authors refer to a paper describing the high sensitivity of the measured ductility to the selected gauge length – this was indeed my point. The authors should go a little further than just rebut my comment and say that this is a known fact. They should comment on this aspect in the paper (or in the supplementary). We are not talking here about small changes. The differences can be a factor of 2 or more ! As a matter of fact, to be accurate, the point is not on the absolute value of the gauge length but on the ratio of the gauge length to the width of the specimen. This is the factor which controls the homogeneity of the stress state inside the gauge section.

3. In the answer “A3”, the authors start with “I agree with the reviewer xxx”. I just want to be sure that the response has been prepared in interaction with all co-authors and endorsed by them – and not only by one of them.

The additional data that are provided (also when answering the comments of the other Referees) are certainly interesting and useful to increase the strength of the paper.

Reviewer #2 (Remarks to the Author):

The authors have carried out additional experiments to address the concerns raised by the reviewer. The reviewer is satisfied with the answers and recommends to publish this paper in Nature Communications.

Reviewer #3 (Remarks to the Author):

The authors addressed all the comments and suggestions. The manuscript is now suitable for publication.

YONSEI UNIVERSITY

Department of Materials Science and Engineering

July 27, 2017

Dear Editor,

We thoroughly revised our manuscript (NCOMMS-17-00882A), titled “Honey-like steel: 1300% superplasticity in a Fe-Mn alloy” to *Nature Communications* based on the reviewer’s comments. The answers to three reviewer’s comments are as follows.

Reviewer #1

Overall comments) The revision has allowed the authors to clarify several important points. They have convincingly answered to my remarks, except perhaps for a few points which require minor revision.

Q1) Regarding the “breakthrough–originality” of the work. My point was to support the fact that reaching 1300% ductility is a breakthrough. The authors answer that the key element is to have a compositionally lean superplastic material more than 1300% ductility. Fine, I’m ready to follow this argument, certainly, and to recognize that the achievement is very significant under this view. But then, the title should be rephrased – because the title makes the reader believe that the focal point is the super high ductility. Perhaps just add the word “lean” in the title “xxx in a lean Fe-Mn alloy” or something like that.

A1) We totally agree with this comment. Thus, we changed the original title “Honey-like steel: 1300% superplasticity in a Fe-Mn alloy” to “**Superplasticity in a lean Fe-Mn-Al steel**” in the revised manuscript.

Q2) Regarding the question of the gauge length which has been selected to perform the tests, the authors refer to a paper describing the high sensitivity of the measured ductility to the selected gauge length – this was indeed my point. The authors should go a little further than just rebut my comment and say that this is a known fact. They should comment on this aspect in the paper (or in the supplementary). We are not talking here about small changes. The differences can be a factor of 2 or more! As a matter of fact, to be accurate, the point is not on the absolute value of the gauge length but on the ratio of the gauge length to the width of the specimen. This is the factor which controls the homogeneity of the stress state inside the gauge section.

A2) We agree with the reviewer’s comment that when the ratio of gauge length to width decreases, inhomogeneous deformation is enhanced, resulting in the so-called draw-in phenomenon and in large elongation. The ratio of gauge length to width in our specimens is 0.5, which is lower than that (0.74-2) in the specimens used in literature [Refs. 6, 8, 13, 16, 17]. Accordingly, if the ratio of gauge length to width increases, total elongation of the specimens used in the present study is expected to be reduced. Therefore, we honestly mentioned this fact in the result part of the revised manuscript as follows.

Revised: However, elongation depends not only on the m value but also on the ratio of gauge length to width ²⁸; the lower the ratio the larger elongation. The ratio of gauge length to width in the specimens used in the present study is 0.5, which is lower than the ratio of gauge length to width (0.74-2) in the specimens used in literature ^{6,8,13,16,17}. Accordingly, if the ratio of gauge length to width increases, the elongation of present specimens is expected to be reduced.

Q3) In the answer “A3”, the authors start with “I agree with the reviewer xxx”. I just want to be sure that the response has been prepared in interaction with all co-authors and endorsed by them – and not only by one of them. The additional data that are provided (also when answering the comments of the other Referees) are certainly interesting and useful to increase the strength of the paper.

A3) The response to the reviewers’ comments was prepared in interaction with all co-authors. At the beginning of the answer “A3”, we mistakenly wrote “I agree ~”.

Reviewer #2

Overall comments) The authors have carried out additional experiments to address the concerns raised by the reviewer. The reviewer is satisfied with the answers and recommends to publish this paper in Nature Communications.

Reviewer #3

Overall comments) The authors addressed all the comments and suggestions. The manuscript is now suitable for publication.

Thank you very much for your kind review.

Sincerely yours,

Prof. Young-Kook Lee

E-mail: yklee@yonsei.ac.kr